# Isoquercitrin from *Apocynum venetum* L. Exerts Antiaging Effects on Yeasts via Stress Resistance Improvement and Mitophagy Induction through the Sch9/Rim15/Msn Signaling Pathway

**DOI:** 10.3390/antiox12111939

**Published:** 2023-10-31

**Authors:** Yanan Liu, Le Shen, Akira Matsuura, Lan Xiang, Jianhua Qi

**Affiliations:** 1College of Pharmaceutical Sciences, Zhejiang University, Yu Hang Tang Road 866, Hangzhou 310058, China; liuyanan1231@zju.edu.cn (Y.L.); 3170101957@zju.edu.cn (L.S.); 2Department of Biology, Graduate School of Science, Chiba University, Chiba 263-8522, Japan; amatsuur@faculty.chiba-u.jp; 3Jinhua Institute of Zhejiang University, Jinhua 321299, China

**Keywords:** isoquercitrin, antiaging, antioxidative stress, anti-thermal stress, mitophagy, Sch9/Rim15/Msn

## Abstract

Background: With the development of an aging sociality, aging-related diseases, such as Alzheimer’s disease, cardiovascular disease, and diabetes, are dramatically increasing. To find small molecules from natural products that can prevent the aging of human beings and the occurrence of these diseases, we used the lifespan assay of yeast as a bioassay system to screen an antiaging substance. Isoquercitrin (IQ), an antiaging substance, was isolated from *Apocynum venetum* L., an herbal tea commonly consumed in Xinjiang, China. Aim of the Study: In the present study, we utilized molecular-biology technology to clarify the mechanism of action of IQ. Methods: The replicative lifespans of K6001 yeasts and the chronological lifespans of YOM36 yeasts were used to screen and confirm the antiaging effect of IQ. Furthermore, the reactive oxygen species (ROS) and malondialdehyde (MDA) assay, the survival assay of yeast under stresses, real-time polymerase chain reaction (RT-PCR) and Western blotting analyses, the replicative-lifespan assay of mutants, such as Δ*sod1*, Δ*sod2*, Δ*gpx*, Δ*cat*, Δ*skn7*, Δ*uth1*, Δ*atg32*, Δ*atg2*, and Δ*rim15* of K6001, autophagy flux analysis, and a lifespan assay of K6001 yeast after giving a mitophagy inhibitor and activator were performed. Results: IQ extended the replicative lifespans of the K6001 yeasts and the chronological lifespans of the YOM36 yeasts. Furthermore, the reactive nitrogen species (RNS) showed no change during the growth phase but significantly decreased in the stationary phase after treatment with IQ. The survival rates of the yeasts under oxidative- and thermal-stress conditions improved upon IQ treatment, and thermal stress was alleviated by the increasing superoxide dismutase (Sod) activity. Additionally, IQ decreased the ROS and MDA of the yeast while increasing the activity of antioxidant enzymes. However, it could not prolong the replicative lifespans of Δ*sod1*, Δ*sod2*, Δ*gpx*, Δ*cat*, Δ*skn7*, and Δ*uth1* of K6001. IQ significantly increased autophagy and mitophagy induction, the presence of free green fluorescent protein (GFP) in the cytoplasm, and ubiquitination in the mitochondria of the YOM38 yeasts at the protein level. IQ did not prolong the replicative lifespans of Δ*atg2* and Δ*atg32* of K6001. Moreover, IQ treatment led to a decrease in Sch9 at the protein level and an increase in the nuclear translocation of Rim15 and Msn2. Conclusions: These results indicated that the Sch9/Rim15/Msn signaling pathway, as well as antioxidative stress, anti-thermal stress, and autophagy, were involved in the antiaging effects of IQ in the yeasts.

## 1. Introduction

The global demographics are swiftly changing towards an aging society, as there is an increasing number of older adults and a decreasing number of young adults available to support them [1]. The decline in physical activity and excessive sedentary behavior among the elderly contribute to the development of various age-related diseases, such as Alzheimer’s disease, cerebral arteriosclerosis, hypertension, and malignant tumors. This present situation imposes a serious challenge to public health and socioeconomics [2]. Currently, numerous potential antiaging molecules have shown promising results, but there is no FDA-approved commercial drug that is capable of delaying aging. As a result, the need for identifying natural compounds with antiaging effects to prevent and treat aging and age-related diseases is exceptionally urgent.

The anti-stress ability of cells weakens with aging, which is the main cause of aging and the development of many age-related diseases [3]. Excessive reactive oxygen species (ROS) and reactive nitrogen species (RNS) can damage the nerve membrane and cause oxidative stress and nitrosative stress. Nitrification stress refers to the biochemical reaction that occurs in conjunction with RNS derived from nitric oxide (NO) and ROS, causing the hydroxylation of the aromatic rings of amino acid residues, which can lead to cell damage or apoptosis and various toxic effects in cells [4]. They play important roles in signaling transduction and inducing the damage of biomacromolecules, such as proteins, nucleic acids, fatty acids, and lipids [4]. The resistance to oxidative, thermal, and osmotic stresses is crucial for extending the lifespans of organisms [5]. Recently, some evidence has reported that traditional Chinese medicine can enhance resistance against oxidative and thermal stress to promote longevity in *Caenorhabditis elegans* [6]. Anti-thermal-stress-related proteins have the ability to promote redox homeostasis by regulating the activity, expression, and degradation of antioxidant enzymes. This ability establishes connections between different stress-resistance effects [7]. Thus, we explore the anti-thermal-stress effect of natural products in this study.

Autophagy plays an important role in aging. There is macro-autophagy, micro-autophagy, and chaperone-mediated autophagy (CMA). Mitophagy is a type of CMA that selectively breaks down the aging or damaged mitochondria [8]. Among the autophagy-related (Atg) proteins, the Atg8 conjugation system is a functional group that plays an important role in the formation and expansion of autophagosomes [9]. Atg32 acts as a mitophagy receptor that interacts with Atg8 to recruit mitochondria to the phagophore assembly site (PAS), the location where autophagosomes are formed [10]. Another important group of proteins in autophagy is the Atg2–Atg18 complex, which is also part of the core Atg proteins and recycles Atg9 from the PAS when autophagy occurs [11]. Therefore, the regulation of autophagy may be an efficient means of antiaging.

The oxidative-stress pathway involves multiple signaling pathways, including the Nrf2-ARE, NF-κB, MAPK, PI3K/AKT, keap1-Nrf2, HIF-1α, AMPK, mTOR, JAK/STAT, TGF-β, and Wnt/β catenin pathways [12]. These pathways collectively regulate biological processes, such as cell growth, reproduction, differentiation, and survival. The mTOR signaling pathway plays a key role in the regulation of protein synthesis, autophagy, and the lifespan of the organism [13]. Rim15, one of the protein kinases, which is located downstream of the TORC1 signaling pathway, is required for cellular protection and chronological-lifespan extension [14]. Sch9 is one key protein to control stress and autophagy via the modification of the Rim15 and Msn2/4 transcription factors [15,16]. The interplay among Sch9, Rim15, and other regulatory factors, such as Msn2/4, affects autophagy and lifespan regulation in response to stress conditions. Understanding these mechanisms can provide insights into the processes of aging and age-related diseases for the development of antiaging drugs.

Yeast models, particularly the K6001 strain and its mutants, are commonly used in antiaging research because of their advantageous characteristics, which are small genomes, short generation times, and low costs [17]. All these properties made yeast a suitable system for the high-throughput drug screening from natural products in our research. The K6001 yeast strain is particularly useful for replicative-lifespan assays. When cultured in glucose medium, only mother cells can divide into daughter cells, whereas daughter cells cannot reproduce the next generation [18]. This replicative aging phenomenon mimics certain aspects of the cellular aging observed in higher organisms and allows for the evaluation of the lifespan extension or aging-related effects in yeast. YOM36 yeast is a prototrophic strain derived from the standard strain BY4742. The construction of YOM36 yeast was initially aimed at addressing the impact of non-essential amino acids on the yeast chronological lifespan [19].

*Apocynum venetum* L. is one of the traditional herbal medicines, and the leaves of this plant are used as tea in China. It is well known that it reduces blood pressure and blood lipids, provides sedation, and has antiaging properties [20]. Isoquercitrin (IQ) from *A. venetum* L. is a naturally occurring substance with antioxidant, anti-proliferative, anti-inflammatory, autophagy-induction, anticancer, and anti-obesity effects [21,22,23]. IQ belongs to the flavonoid-type molecule possessing four phenolic hydroxyl groups at positions C-5, C-7, C-3′, and C-4′, as well as a glucosylation at C-3. It alleviates ethanol-induced liver toxicity, oxidative stress, and inflammatory responses through the Nrf2/ARE antioxidant signaling pathway [24]. Moreover, IQ regulates the nuclear factor-κB (NF-κB) transcriptional regulatory system to regulate the expression of nitric oxide synthase 2 (NOS2) [25]. However, the antiaging effect of IQ is rarely mentioned, and the action mechanism of the antiaging effect exerted by IQ remains unclear. On the basis of previous research, the stress-resistance and autophagy-induction abilities of IQ were chosen to elaborate the mechanism underlying its antiaging effect in yeasts. Here, we report that IQ exerts an antiaging effect on yeasts via improving stress resistance and inducing mitophagy through the Sch9/Rim15/Msn signaling pathway.

## 2. Materials and Methods

### 2.1. General

Analytically pure reagents (chloroform, isopropanol, and ethanol from Sinopharm Chemical Reagent Co., Ltd., Shanghai, China) were used in this study. The following compounds and reagents were purchased from the indicated companies: resveratrol (RES) (J&K Scientific Ltd., Beijing, China); rapamycin (Solarbio, Beijing, China); wortmannin (Beyotime Biotechnology, Shanghai, China); CSK3-IN-3 (HY-153089, MedChemExpress, Shanghai, China); Mdivi-1 (HY-15886, MedChemExpress, Shanghai, China); dimethyl sulfoxide (DMSO) (Sigma, Saint Louis, MO, USA); Hoechst 33,342 (Macklin, Shanghai, China); and 4′,6-diamidino-2-phenylindole (DAPI) dihydrochloride (Macklin, Shanghai, China). DMSO was employed as both a solvent for dissolving compounds and as the negative control group in the yeast activity evaluation system.

### 2.2. Isolation and Purification of Isoquercitrin

The process of preparing the IQ from *A. venetum* L. was described in a previous study [23], and the chemical structure of IQ is shown in Figure 1a.

### 2.3. Yeast Strains, Culture Medium, and Lifespan Assay

In this study, the K6001 yeasts derived from W303 and the YOM36 yeasts derived from BY4742 were used in the lifespan assay. The Δ*sod1*, Δ*sod2*, Δ*uth1*, Δ*skn7*, Δ*gpx*, Δ*cat*, Δ*atg2*, Δ*atg32*, and Δ*rim15* yeast strains with the K6001 background, BY4741 yeast strain, YOM38 yeast strain containing pRS316-*GFP-ATG8* plasmid, and BY4741 yeast strains containing *sfGFP-Sch9-5HA::LEU2*, *Rim15-GFP::his3MX*, and *Msn2-GFP::kanMX* plasmids were applied in the mechanism-of-action analysis. The genotypes of the yeast strains are listed in Appendix A, as described in a previous study [26].

The lifespan assay was conducted according to previous research [26]. In the present study, the replicative lifespan of K6001 yeast was applied to screen the antiaging effect of IQ at 0.3, 1, 3, 10, 30, and 100 μM, and the chronological lifespan of YOM36 yeast was used to confirm the antiaging effect of IQ at 1, 10, and 30 μM. The replicative-lifespan assay of the Δ*gpx*, Δ*cat*, Δ*sod1*, Δ*sod2*, Δ*uth1*, Δ*skn7*, Δ*atg2*, Δ*atg32*, and Δ*rim15* yeast strains with the K6001 background was similar to that of the K6001. The details of the lifespan assay are shown in Appendix A.

### 2.4. Growth Curve Determination

First, the YOM36 yeast cells with an initial optical density at 600 nm (OD_600_) of 0.01 were cultured in 100 mL of synthetic defined (SD) medium, containing IQ (0, 1, 10, and 30 μM) or rapamycin (1 μM). At 1, 1.5, 2, 2.5, 3, 3.5, 4, 5, 5.5, 6.5, 7.5, 8.5, 9.5, and 10.5 days, the OD_600_ values of the yeasts were measured by using the Eppendorf Biophotometer Plus (Eppendorf Company, Hamburg, Germany).

### 2.5. Determination of Yeast Survival Ability under H_2_O_2_ Stimulation

First, BY4741 yeasts were cultured in yeast extract peptone dextrose (YPD) for 24 h with shaking. On day 2, the yeasts were treated with compounds at an initial OD_600_ value of 0.1. After 24 h, the yeast suspension was diluted to an OD_600_ value of 2. About 5 μL of yeast broth was dropped on YPD agar plates containing 11 mM H_2_O_2_; one optimum concentration of H_2_O_2_ was applied, referring to our previous research [27]. The growth of the yeast cells was photographed after incubating at 28 °C for 2 days. In the quantitative assay, a total of 200 yeasts from each group were spread on YPD agar plates with or without 5.5 mM H_2_O_2_. One optimum concentration of H_2_O_2_ was used, based on our previous study [27], and the plates were cultured at 28 °C for 48 h. The colony-forming units (CFUs) formed on each plate were counted to evaluate the antioxidative-stress activity, and the survival rate was defined as the ratio of the number of CFUs surviving with H_2_O_2_ at 5.5 mM divided by the number of CFUs formed without H_2_O_2_.

### 2.6. Thermal-Stress-Resistance Assay

The anti-thermal-stress assay was conducted according to another report [28]. First, we conducted an initial experiment. In brief, the YOM36 yeast was cultured in YPD for 12 h (180 rpm, 28 °C). The yeast culture broth was diluted into five different concentrations by 10 times the gradient dilution. About 5 μL of cultured yeast with different concentrations was dropped on the YPD agar plates. For the quantitative anti-thermal-stress assay, a total of 200 yeasts were spread on each YPD agar plate. The plates were incubated at 28 °C for 24–48 h with or without preheating at 55 °C or 60 °C for a certain time period. After determining the optimal thermal-stress condition, the YOM36 yeasts were treated with IQ at 0, 1, 10, and 30 μM or RES at 10 μM with an initial OD_600_ value of 0.1. After incubation for 24 h, the yeasts were handled following the method described above for the qualitative or quantitative analysis of the anti-thermal stress. The anti-thermal-stress assay of the K6001 and Δ*sod1*, Δ*sod2*, and Δ*rim15* yeasts with the K6001 background was the same as that of YOM36.

### 2.7. Measurement of ROS, RNS, and MDA Levels of Yeasts

For the ROS assay, BY4741 yeasts were cultured in YPD liquid medium for 24 h with shaking. On the next day, the BY4741 yeasts were cultured in YPD containing compounds with an initial OD_600_ value of 0.1. After 23 h, about 1 mL of yeast suspension was harvested and treated with 2′,7′-dichlorodihydrofluorescein diacetate (DCFH-DA) with a final concentration of 10 μM and then incubated with shaking for 60 min at 28 °C under dark conditions. Finally, the yeasts were washed three times with phosphate-buffered solution (PBS), and the DCF (2′,7′-dichlorofluorescein) fluorescence intensity of the yeasts was measured with excitation and emission wavelengths of 488 and 525 nm, respectively, by using a Spark microplate reader (Tecan Trading Co., Ltd., Männedorf, Switzerland).

For the RNS measurement, YOM36 yeasts were gathered and ground with a grinder (Jingxin Industrial Development Co., Ltd., Shanghai, China) at 70 Hz for 1 min after adding grinding beads. The amounts of yeasts in the rapamycin-treated group were too low to extract protein at day 1, so the RNS measurement of the rapamycin-treated group was skipped at this time point. Then, the cell lysates were centrifuged (4 °C, 12,000 rpm, 10 min) to obtain the supernatant as protein samples to evaluate the RNS level following the instructions of the RNS assay kit (Bestbio Biotechnology Company, Nanjing, China). The details of the assay are shown in Appendix A.

For the malondialdehyde (MDA) assay, BY4741 yeasts were cultured in YPD containing compounds with an initial OD_600_ value of 0.1 for 24 h. The yeasts were gathered, washed with PBS three times, and ground with a grinder at 70 Hz for 1 min. Then, the cell lysates were centrifuged to obtain the supernatant to evaluate the MDA by using the MDA assay kit (Nanjing Jiancheng Institute, Nanjing, China). The details of the assay are shown in Appendix A.

For the ROS and MDA assays under thermal-stress conditions, the YOM36 yeasts that were incubated with IQ or RES for 24 h were first heated at 60 °C for 30 min. Then, the corresponding assay was conducted following the method described above.

### 2.8. Antioxidant Enzyme Activity Determination

The BY4741 yeasts were cultured and incubated with compounds as described for the MDA assay. The yeasts were washed with PBS three times and ground with a grinder at 70 Hz for 1 min. Then, the protein sample was obtained after centrifugation (12,000 rpm, 10 min) at 4 °C. According to a previous study [26], the protein sample was diluted to an appropriate concentration to measure the superoxide dismutase (Sod), catalase (Cat), and glutathione peroxidase (Gpx) activities by using Sod (Nanjing Jiancheng Institute, Nanjing, China), Cat, and Gpx (Beyotime Limited Company, Shanghai, China) assay kits, respectively. For the Sod assay under thermal-stress conditions, the YOM36 yeasts that were incubated with IQ or RES for 24 h were first heated at 60 °C for 30 min. The details of the assay are shown in Appendix A.

### 2.9. Real-Time Polymerase Chain Reaction (RT-PCR) Analysis

The cDNA samples of yeast were prepared as described in a previous study [26]. The details of the assay are shown in Appendix A. The primers of *MSN2*, *MSN4*, and *TUB1* are shown in Appendix A. The thermal-recycling parameters for the RT-PCR were as follows: *MSN2* and *MSN4*, 95 °C for 2 min, followed by 40 cycles, 95 °C for 15 s, 55 °C for 20 s, and 70 °C for 20 s. All results were standardized to *TUB1* levels, and the relative mRNA transcript levels were analyzed using the 2^−ΔΔCt^ formula.

### 2.10. Observation of Autophagy and Mitophagy in Yeasts

The autophagy assay was conducted according to a previous study [26]. In short, YOM38 yeasts were cultured in YPD with shaking under dark conditions. After 24 h, the yeasts were diverted to an SD medium and treated with IQ (0, 1, 10, and 30 μM) with or without wortmannin (200 nM) for 22 h. Resveratrol (300 μM) was used as the positive control. After 22 h, the appropriate number of yeasts were collected, stained with DAPI (final concentration 1 μg/mL) for 8 min in the dark, and washed three times with PBS. Yeasts were finally suspended in 30% glycerin solution and photographed with a confocal fluorescence microscope (Olympus FV1000BX-51, Tokyo, Japan). To measure the mitophagy, the yeasts were firstly stained with 300 nM MitoTracker Red CMXRos (Beyotime, Shanghai, China) at 37 °C under dark conditions for 60 min before staining with DAPI (final concentration 1 μg/mL). The percentage of yeasts with free green fluorescent protein (GFP) and the colocalization of free GFP and MitoTracker Red were obtained to evaluate the ability of IQ to induce autophagy or mitophagy. Furthermore, we detected whether the mitophagy inhibitor Mdivi-1 and activator GSK3-IN-3 affected the replicative lifespan of the K6001 yeast. The replicative-lifespan assay of K6001 referenced Appendix A.

To detect whether Rim15 is involved in the induced autophagy of IQ, the K6001 and Δ*rim15* of the K6001 yeasts were cultured in galactose liquid medium for 24 h with shaking. On day 2, the K6001 or Δ*rim15* of the K6001 yeasts with an initial OD_600_ value of 0.1 were co-incubated with IQ at 10 μM or RES at 300 μM. After 22 h of cultivation, the yeasts were gathered and stained with an autophagy detection kit (Enzo Life Sciences, Inc., New York, NY, USA) according to the instructions. The green dye was diluted at a ratio of 4:1000. The yeasts were stained with the green dye under dark conditions at 37 °C for 1 h. Then, the yeasts were washed with PBS and stained with DAPI (final concentration: 1 μg/mL) for 8 min under dark conditions before we observed them with a confocal fluorescence microscope.

### 2.11. Observation of the Nuclear Translocation of GFP-Rim15 and GFP-Msn2 in Yeasts

In brief, an appropriate amount of BY4741 yeasts containing *Rim15-GFP::his3MX* and *Msn2-GFP::kanMX* plasmids were cultured in YPD with shaking under dark conditions. After 10 h, the BY4741 yeasts containing the *Msn2-GFP::kanMX* plasmid were treated with IQ at 0, 1, 10, and 30 μM or rapamycin at 1 μM for another 2 h. For BY4741 yeasts containing the *Rim15-GFP::his3MX* plasmid, the cultivation duration was 3 days before 6 h of treatment with the compounds. After incubating for a corresponding time, the BY4741 yeasts were harvested and stained with Hoechst 33,342 (final concentration 1 μg/mL) for 8 min under dark conditions and were then washed three times with PBS. The yeasts were finally suspended in a 30% glycerin solution and photographed with a fluorescence microscope.

### 2.12. Western Blot Analysis

For the free GFP and ubiquitin, the YOM38 yeasts were treated with IQ (0, 1, 10, and 30 μM) with or without wortmannin (200 nM) for 22 h. Resveratrol (300 μM) was used as the positive control. The harvested yeasts were ground with a grinder at 70 Hz for 1 min and centrifuged (12,000 rpm, 10 min) to obtain the protein samples for the Western blot analysis. Mitochondrial protein was obtained followed the description in a previous study [26]. First, the lysate was centrifuged two times (5000 rpm, 15 min). The mitochondrial pellet was obtained after centrifuging the supernatants (12,000 rpm, 30 min). Then, the precipitate was lysed with RAPI lysis buffer (CoWin Biotech, Beijing, China) containing a 1% protease inhibitor cocktail (CoWin Biotech, Beijing, China) for 15 min. Supernatants were obtained as protein samples to evaluate the ubiquitin expression in the mitochondria after centrifugation (12,000 rpm, 15 min). The polyvinylidene fluoride (PVDF) membranes were incubated with primary antibodies against GFP (^#^598, Medical & Biological Laboratories, Nagoya, Japan), β-actin (^#^db6010, Diagbio Biotech, Hangzhou, China), ubiquitin (^#^3933, Cell Signalling Technology, Boston, MA, USA), and mitochondrial outer membrane protein porin 1 (VDAC1) (^#^ab110326, Abcam Trading Company Ltd., Cambridge, UK). The secondary antibodies horseradish peroxidase-linked goat anti-rabbit (^#^CW0103, CoWin Biotech, Beijing, China) for the primary antibodies of GFP and ubiquitin and horseradish peroxidase-linked goat anti-mouse IgGs (^#^CW0102, CoWin Biotech, Beijing, China) for the primary antibodies of β-actin and VDAC1 were used in this study. Protein bands were obtained via exposure via a Bio-Rad chemiluminescence imager (Bio-Rad Laboratories, Hercules, CA, USA). ImageJ software (Version 1.42q National Institute of Health, Rockville, MD, USA) was used to digitalize the bands.

For the sfGFP-Sch9-5HA, the BY4741 yeast containing the *sfGFP-Sch9-5HA::LEU2* plasmid was cultured in YPD with shaking under dark conditions. After 12 h, the cells were treated with Rap (1 μM) or IQ (1, 10, and 30 µM) for 2 h. The harvested yeasts were finally suspended in 500 μL of RAPI lysis buffer containing a 1% protease inhibitor cocktail. The protein samples for the Western blotting were obtained following the method mentioned above. The primary antibodies against HA (^#^16B12, BioLegend, San Diego, CA, USA) and β-actin (^#^db6010, Diagbio Biotech, Hangzhou, China) and the secondary antibody horseradish peroxidase-linked goat anti-mouse IgGs (^#^CW0102, CoWin Biotech, Beijing, China) were used in this study.

### 2.13. Statistical Analysis

Experiments were repeated three times, and the data are presented as means ± SEMs. Significant differences among the groups in all experiments were evaluated via one-way ANOVA, followed by Tukey’s Multiple Comparison Test, on GraphPad Prism software (Version 9.0.0 (121), GraphPad Prism, San Diego, CA, USA). The chronological-lifespan assay of the yeast was analyzed via the log-rank (Mantel–Cox) test. Statistical significance was represented as *p* < 0.05.

## 3. Results

### 3.1. IQ Extends the Lifespans of Yeasts

K6001 is a yeast strain with the W303 background, with the characteristic that only the mother cell with a specific expression of *HO::CDC6* can generate offspring on glucose agar plates [18]. In this study, the antiaging effect of IQ at 0, 0.3, 1, 3, 10, 30, and 100 μM was evaluated via the K6001 replicative-lifespan bioassay system (Appendix A). In the replicative-lifespan assay, the average lifetimes of the effective-concentration groups were as follows: 8.05 ± 0.47 generations in the negative control group; 11.63 ± 0.55 (*p* < 0.001) in the RES-treated group at 10 μM; and 9.88 ± 0.73 (*p* < 0.05), 10.55 ± 0.57 (*p* < 0.01), and 11.93 ± 0.60 (*p* < 0.001) generations in the IQ-treated groups at 1, 10, and 30 μM (Figure 1b). The chronological lifespan of YOM36 after treatment with IQ was then determined to confirm the antiaging potential. In the chronological-lifespan assay, the median survival of the IQ-treated group at 30 μM was 11 days, which was obviously longer than that of the control group (9 days) (Figure 1c). The IQ prolonged the replicative lifespan of K6001, as well as the chronological lifespan of YOM36. These results generally indicated the significant antiaging effects of IQ on yeasts.

### 3.2. The Effect of IQ on RNS of Yeasts during Chronological Aging

In addition to its important signaling transduction role, excessive RNS can damage intracellular components, thereby promoting nitrosative stress [4]. The change in RNS at different growth phases during chronological aging was measured to explore whether IQ prolongs the chronological lifespans of yeasts through antioxidative stress. As Figure 2a shows, the yeasts were in log-phase growth within 2 days in the IQ-treated and control groups; however, for the rapamycin-treated group, the corresponding time period was within 3.5 days. The growth rate of yeast slowed down from 2 days until entering the stationary phase at 5 days in the IQ-treated and control groups. The amounts of yeasts in the rapamycin-treated group were too low to extract protein at day 1, so the RNS measurement of the rapamycin-treated group was skipped at this time point.

As shown in Figure 2b–d, the level of RNS in the growth phase was not affected after treatment with IQ compared with that of the control group. However, in the rapamycin-treated group, the RNS level was obviously increased at 2 d and 3.5 days. The IQ significantly reduced the RNS levels compared with the control group at 1 (*p* < 0.001), 10 (*p* < 0.001), and 30 μM (*p* < 0.001) at 5 days (Figure 2e). As a result of the inhibitory effect of rapamycin on the growth of yeasts, at the same time point, the influence on the RNS levels was different between the rapamycin and IQ due to the different growth phases (Figure 2e). These data indicated that the IQ decreased RNS in the early stationary phase during the chronological aging to extend the chronological lifespans of the yeasts.

### 3.3. IQ Increases the Antioxidative-Stress Activity of Yeasts

Oxidative stress is a key factor leading to aging and neurodegenerative diseases, and multiple natural products have significantly prolonged the lifespans of model organisms via antioxidative stress [6,26,27]. Therefore, the survival rate of the BY4741 yeast under H_2_O_2_ stimulation was examined to clarify whether antioxidative stress is involved in the antiaging mechanism of IQ. The results of the qualitative assay are shown in Figure 3a. The BY4741 yeasts that were treated with IQ at 10 and 30 μM grew better on a glucose agar plate containing 11 mM of H_2_O_2_ compared with the negative control. Moreover, the survival rate of the yeast under H_2_O_2_ stimulation after treatment with IQ was quantitatively confirmed on the glucose agar plate with or without 5.5 mM of H_2_O_2_. The survival rates of each group were as follows: 52.0 ± 5.6% for the negative control; 80.8 ± 1.5% (*p* < 0.001) for the RES-treated group at 10 μM; 56.1 ± 4.8%, 90.8 ± 2.9% (*p* < 0.001), and 96.8 ± 0.8% (*p* < 0.001) in the IQ-treated groups at 1, 10, and 30 μM, respectively (Figure 3b). These results implied that the beneficial effects of IQ on the yeast lifespan were preliminarily attributed to its antioxidative-stress activity.

Low concentrations of ROS can maintain normal cell function, but abnormally high concentrations beyond the elimination limit of the antioxidant system in vivo may cause damage to growth factors, transcription factors, proteins, nucleic acids, and lipids [4]. MDA is the end product of membrane lipid peroxidation caused by ROS [29]. Therefore, the ROS and MDA levels were investigated to determine the extent of the intracellular oxidation after IQ treatment. Figure 3c shows that IQ significantly decreased the ROS levels from 1016.2 ± 62.6 to 769.9 ± 16.3, 545.2 ± 57.2, (*p* < 0.01), and 500.3 ± 105.1 (*p* < 0.01) at 1, 10, and 30 μM, respectively. The positive-control RES at 10 μM decreased the ROS from 1016.2 ± 62.6 to 603.7 ± 52.1 (*p* < 0.05). Figure 3d implies that the MDA decreased from 0.62 ± 0.03 to 0.50 ± 0.03 (*p* < 0.05), 0.52 ± 0.02 (*p* < 0.05), 0.50 ± 0.04 (*p* < 0.05), and 0.49 ± 0.02 (*p* < 0.01) after treatment with RES at 10 μM and IQ at 1, 10, and 30 μM, respectively. Therefore, IQ could effectively reduce the ROS and MDA levels of the BY4741 yeast, which further demonstrated that antioxidative stress played a key role in the antiaging effect of IQ.

Sod, Cat, and Gpx constitute an enzymatic antioxidant system and play a vital role in the balance of oxidation and antioxidative stress [30]. Therefore, the activities of these antioxidant enzymes under physiological conditions were assessed to clarify the molecular mechanism of the antioxidative-stress effect exerted by IQ. Figure 3e,f indicate that the total Sod activity was obviously enhanced after treatment with IQ at 1 (*p* < 0.05), 10 (*p* < 0.05), and 30 μM (*p* < 0.001), whereas the Sod1 enzyme activity was improved after treatment with IQ at 1 (*p* < 0.05), 10 (*p* < 0.001), and 30 μM (*p* < 0.001), respectively. Meanwhile, increased Cat enzyme activity was observed in the IQ-treated group at 1 (*p* < 0.05), 10 (*p* < 0.01), and 30 μM (*p* < 0.01), whereas the Gpx enzyme activity was evidently enhanced upon IQ treatment at 10 (*p* < 0.05) and 30 μM (*p* < 0.05) (Figure 3g,h). These data indicated that the IQ countered oxidative stress by increasing the activities of Sod, Cat, and Gpx, thereby completing the antiaging effect.

### 3.4. IQ Confers Thermotolerance on Yeasts via SOD

Natural products that increase the cell resistance to thermal stresses have been shown to decelerate the aging process and extend longevity in yeast [28]. Therefore, we investigated the effect of IQ on the susceptibility of YOM36 yeasts to thermal stress. Compared with the yeasts that were not heated, the survival ability of the yeasts that experienced thermal stress at 60 °C for 30 min was obviously declined (Figure 4a). Therefore, heating at 60 °C for 30 min was chosen as the condition for the formal thermal-stress examination. The YOM36 yeasts at 1 × 10^5^ times dilution grew better after IQ treatment at 1, 10, and 30 μM for 24 h followed by heating at 60 °C for 30 min compared with the control group (Figure 4b). As Figure 4c shows, different from the qualitative assay, heating at 60 °C for 25 min led to a 50% survival rate, and this was selected as the condition for the formal experiment in the quantitative assay. Figure 4d implies that IQ at 10 and 30 μM made the yeasts more resistant to thermal stress and increased the survival rates of the yeasts from 51.7 ± 2.3% to 76.7 ± 2.9% (*p* < 0.05) and 97.0 ± 5.7% (*p* < 0.01), respectively.

Multiple-stress resistance is necessary for lifespan extension, and antioxidants can protect *C. elegans* against thermal stress [6]. The ROS, MDA, and Sod levels after thermal stimulation were measured to clarify whether there is crosstalk between the antioxidative-stress and anti-thermal-stress effects exerted by IQ. As Figure 4e,f display, IQ could effectively reduce the ROS at 1 (*p* < 0.05), 10 (*p* < 0.01), and 30 μM (*p* < 0.01) and the MDA levels at 1, 10, and 30 μM (*p* < 0.05) under room-temperature conditions. The ROS and MDA levels were significantly improved after heating at 60 °C for 30 min (*p* < 0.001). Unexpectedly, the IQ failed to decrease the ROS and MDA levels under thermal-stress conditions (Figure 4e,f). Furthermore, the T-Sod and CuZn-Sod activities after IQ treatment under thermal stress were explored. Heating at 60 °C for 30 min resulted in a decrease in the Sod enzyme activities, and IQ could increase the T-Sod and CuZn-Sod activities under room-temperature or thermal-stress conditions (Figure 4g,h). These results indicated that IQ exerted anti-thermal-stress effects, and it enhanced Sod enzyme activities under thermal stress.

Under thermal stress, the expression levels of *SOD1* and *SOD2* were significantly upregulated, exerting a heat-tolerance effect [31]. Anti-thermal-stress assays of K6001, Δ*sod1*, and Δ*sod2* with the K6001 background after IQ treatment were conducted to explore whether *SOD1* and *SOD2* participate in the anti-heat-stress effect of IQ. Figure 4i implies that heating at 55 °C for 30 min reduced the survival ability of the K6001 yeasts in the control group compared with 28 °C. IQ treatment improved the heat tolerance of K6001 yeasts under heat stress. Interestingly, IQ failed to increase the survival abilities of Δ*sod1* and Δ*sod2* at 55 °C compared with the wild type. These results confirmed that *SOD1* and *SOD2* were involved in the anti-thermal-stress effects of the IQ.

### 3.5. SOD1, SOD2, CAT, GPx, SKN7, and UTH1 Are Involved in the Antiaging Effect of IQ

To further confirm the relationship between the antioxidant stress effect of IQ and its antiaging function, the Δ*sod1*, Δ*sod2*, Δ*cat*, Δ*gpx*, Δ*skn7*, and Δ*uth1* of K6001 yeasts were employed in the replicative-lifespan assay. As Figure 5a–f display, the average generation of Δ*uth1* was longer than that of the K6001 wild type, whereas the average lifespans of Δ*sod1*, Δ*sod2*, Δ*cat*, Δ*gpx*, and Δ*skn7* were similar to that of the K6001 wild strain. Furthermore, IQ failed to prolong the average lifetimes of Δ*sod1*, Δ*sod2*, Δ*cat*, Δ*gpx*, Δ*skn7*, and Δ*uth1* compared with the negative control. These results further confirmed that the *SOD1*, *SOD2*, *CAT*, *GPx*, *SKN7*, and *UTH1* genes were involved in the lifetime prolongation effect of the IQ. The average replicative lifespans of the K6001 yeast and its mutants are shown in Appendix A.

### 3.6. Effects of IQ on Autophagy and Mitophagy in YOM38 Yeasts

To date, about 40 Atg proteins have been identified in *Saccharomyces cerevisiae*. Atg8 is involved in the elongation and closure of autophagosome membranes and acts as an autophagosome marker [9]. YOM38 yeasts containing the pRS316-*GFP-ATG8* plasmid were applied in the autophagy-induction assay to monitor the autophagy flux, as described in a previous study [26]. Mitophagy is a selective type of autophagy, degrading the damaged mitochondria to maintain cell homeostasis and delay cell senescence and death [8]. MitoTracker Red CMXRos was applied to mark the mitochondria and monitor the mitophagy. Wortmannin is a specific PI3K inhibitor that inhibits the Atg protein core complex and autophagy-specific class III PI3K complex to prevent phagosome formation [32]. Figure 6a–c display that IQ significantly improved the percentage of YOM38 yeasts with free GFP at 1 (*p* < 0.05), 10 (*p* < 0.05), and 30 μM (*p* < 0.01), and the ratio of yeasts with the colocalization of free GFP and MitoTracker Red was obviously increased after treatment with IQ at 10 and 30 μM (*p* < 0.05). Moreover, the induction effect could be inhibited by wortmannin at 200 nM.

The ubiquitin proteasome system is crucial for controlling the degradation of organelles and proteins. Ubiquitin binds to damaged mitochondria and labels them as the substrates for mitophagy [33]. The expression levels of free GFP and ubiquitin at the protein level were analyzed via Western blot to further confirm the autophagy-induction effect of IQ. Figure 6d–f and Appendix A indicate that the IQ significantly increased the expression of free GFP at 1 (*p* < 0.01), 10 (*p* < 0.001), and 30 μM (*p* < 0.001). The increase in free GFP in the yeasts that were co-incubated with IQ at 10 and 30 μM was decreased by wortmannin (*p* < 0.01). Meanwhile, IQ could enhance the expression of ubiquitin, and the increase in ubiquitin via IQ decreased after treatment with wortmannin. Atg2 and Atg32 are critical to mediate the degradation of mitochondria via mitophagy [10]. Therefore, a replicative-lifespan assay of the Δ*atg2* and Δ*atg32* strains with the K6001 background was further conducted. As indicated in Figure 6g,h, the average generations of these two mutants failed to be extended after the IQ treatment, confirming that the *ATG2* and *ATG32* genes were involved in the mitophagy-induction effect of IQ. Furthermore, we used mitophagy inhibitor Mdivi-1 and activator GSK3-IN-3 to check the effect of mitophagy on the lifespan of the K6001 yeast. As expected, Mdivi-1 did not affect the replicative lifespan, and GSK3-IN-3 significantly increased the replicative lifespan of the K6001 yeast at doses of 1, 3, and 10 μM (Figure 6i and Appendix A, *p* < 0.05, *p* < 0.001, and *p* < 0.01, respectively). These data indicated that IQ could significantly induce autophagy and mitophagy to produce the antiaging effect in yeasts.

### 3.7. IQ Inhibits the Expression of Sch9 and Promotes GFP-Rim15 and GFP-Msn2 Transfer into the Nucleus in Yeasts

The downregulation of nutrient signaling through the TORC1/Sch9 pathway significantly extends the chronological lifespan of yeast [13]. Downregulated TORC1/Sch9 signaling leads to the nuclear translocation of Rim15, and the increased nuclear accumulation of Rim15 can prolong the lifetime. Rim15 triggers the anti-stress response for ROS degradation through downstream transcription factors, including Msn2/4 [14]. Msn2/4 can induce cellular protection by enhancing general stress resistance and autophagy [15,16]. The effects of IQ on the expression level and subcellular localization of the TORC1 signaling pathway-related proteins were further explored to clarify its antiaging effect.

The results of the Western blot analysis indicated that IQ effectively inhibited the Sch9 at the protein level after 2 h of treatment at 1, 10, and 30 μM (*p* < 0.001); this result was better than rapamycin at 1 μM (Figure 7a,b and Appendix A). This result revealed that IQ extended the lifespans of the yeasts by inhibiting the expression of Sch9 at the protein level. As Figure 7c indicates, the antiaging effect of IQ was significantly weakened in the Δ*rim15* of K6001, indicating that Rim15 played a key role in the lifespan extension effect of IQ. The average lifespans of K6001 and Δ*rim15* of K6001 are displayed in Appendix A. Subsequently, the intracellular localization of the GFP-Rim15 fusion protein after IQ treatment was examined. Figure 7d,e show that IQ significantly enhanced the nuclear transfer of GFP-Rim15 at 1 (*p* < 0.01), 10 (*p* < 0.001), and 30 μM (*p* < 0.05). These data indicated that Rim15 is involved in the antiaging effect of IQ. Moreover, IQ could effectively improve the gene expression of *MSN2* at 10 μM (*p* < 0.01) and 30 μM (*p* < 0.01), as well as that of *MSN4* at 10 μM (*p* < 0.01), after 24 h (Appendix A). BY4741 yeasts containing the GFP-Msn2 plasmid were employed to observe the subcellular localization of Msn2. IQ obviously promoted the nuclear translocation of GFP-Msn2 at 10 μM (*p* < 0.001) and 30 μM (*p* < 0.01), as rapamycin did at 1 μM (*p* < 0.001) (Figure 7f,g). These results suggested that IQ promoted the nuclear transfer of GFP-Msn2 in the yeast, triggering the transduction of stress-resistance signaling.

IQ inhibited the expression of Sch9 at the protein level and increased the nuclear translocation of GFP-Rim15 and GFP-Msn2, thereby exerting anti-stress and autophagy-induction effects to prolong the lifespans of the yeasts.

### 3.8. IQ Exerts Autophagy-Induction and Thermal-Stress-Resistance Effects via Rim15

Rim15 is required for the induction of autophagy that occurs upon the inhibition of Sch9 [16]. Compared with the K6001 yeast, the rate of autophagosome formation in the negative control and RES- and IQ-treatment groups significantly decreased in the ∆*rim15* of the K6001 yeasts at *p* < 0.05, *p* < 0.01, and *p* < 0.05 (Figure 8a,b). Rim15 is also involved in stress-resistance signals to appropriately induce a transcriptional program that increases survival in G_0_. The regulon downstream of *RIM15* comprises classical anti-stress genes, such as *MSN2* and *MSN4* [15]. A qualitative anti-thermal-stress assay on ∆*rim15* of the K6001 yeasts was conducted to investigate whether *RIM15* is involved in the anti-stress effect exerted by IQ. The Δ*rim15* of the K6001 yeasts exhibited poor heat resistance. Compared with the K6001 yeasts, the anti-thermal-stress effect of the yeasts in the IQ-treated group was significantly weakened in the Δ*rim15* of the K6001 yeasts at 1 × 10^4^ and 1 × 10^5^ dilutions (Figure 8c). These results indicated that Rim15 plays an important role in the autophagy-induction and anti-thermal-stress effects of IQ.

## 4. Discussion

IQ from *A. venetum* L. is a bioactive flavonoid with antioxidative-stress, anti-inflammatory, autophagy-inducing, and anti-obesity properties [21,22,23]. In the present study, we utilized the lifespan assay of yeast to investigate the antiaging effect and mechanism of action of this compound. The changes in the replicative lifespan and chronological lifespan of the yeast in Figure 1b,c indicated that IQ had antiaging effects on the yeast. This result is consistent with our previous studies [26,27]. Recently, it has been indicated that quercetin can clear senescence cells in organisms to produce the antiaging effect [34]. IQ is quercetin-3-glucoside and has better biological activity and bioavailability than quercetin [35]. This gives us inspiration, as IQ also has the same function as quercetin. In the future, we will screen the small molecules from nature products that only kill aging cells and do not affect normal cells. In addition, IQ can inhibit α-glucosidase activity to treat hyperglycemia [36], promote cell apoptosis to inhibit human osteosarcoma cell proliferation [37], and exert an anti-neuroinflammation effect in LPS-induced neuroinflammatory mice [38]. Moreover, IQ is one small-molecule compound with low toxicity and high biological activity. These properties of IQ endow it with the potential to be developed as a promising therapeutic agent for age-related diseases.

In our previous studies, we usually focused on oxidative stress and autophagy to analyze the mechanism of action [26,27]. In this study, we shifted our key point to other stresses, such as nitrosative stress and thermal stress. First, we investigated the changes in the RNS of YOM36 yeast during the chronological-lifespan assay. The results in Figure 2 suggested that RNS and the chronological lifespan were related to each other, and IQ extended the chronological lifespan of YOM36 via a reduction in RNS in the early stationary phase. Interestingly, the RNS levels of the rapamycin-treated group in the growth phase at 2, 3.5, and 5 days increased compared with those of the control group. This result may be why the growth of YOM36 yeast in the growth phase was so slow after the rapamycin treatment. At this point, our results were not consistent with another study, which reported that rapamycin can block the cell cycle progression in the early G1 phase, driving cells into a G0 state [39].

The development of resistance to chronic oxidative and thermal stresses and enhanced autophagy are essential longevity-extending processes in evolutionarily distant organisms, including yeasts and *Caenorhabditis elegans* [5,6,26,27]. Thus, we also investigated the effect of IQ on the oxidative stress, thermal stress, and autophagy. The changes in the ROS and MDA, survival rate under thermal stress, activity of antioxidative enzymes, replicative lifespan of the mutants in Figure 3, Figure 4 and Figure 5, and enhancement on autophagy in Figure 6 proved that stress resistance and autophagy play important roles in the antiaging effect of IQ. Moreover, there is some evidence that the upregulation of *SOD* gene expression exerts heat-tolerance effects under thermal stress [31,40]. To understand whether these two enzymes are involved in the anti-thermal-stress effect of IQ, we measured the Sod enzyme activity and the survival viabilities of Δ*sod1* and Δ*sod2* of K6001 yeast under thermal stress. The results in Figure 4 confirm that *SOD1* was involved in the anti-thermal-stress effects of IQ.

The mTOR signaling pathway is closely related to stress resistance and autophagy. Downregulation of the highly conserved mTOR signal can prolong longevity by increasing stress resistance and autophagy induction [14]. It is worth exploring whether the mTOR signaling pathway is involved in the antiaging effects of IQ. The reduction in Sch9 at the protein level and increase in nuclear translocation of GFP-Rim15 and GFP-Msn2 in Figure 7 clarified that IQ obviously promoted the nuclear translocation of Rim15 and Msn2 to trigger the transcription of anti-stress or autophagy-induction signals. To understand the relationship among Rim15, autophagy, and thermal stress, the *rim15* of K6001 was applied to observe the changes in the IQ effects of autophagy and thermal stress. The results in Figure 8 suggest that Rim15 was a key protein, which controls autophagy and the general stress response, required for IQ to exert autophagy-induction and thermal-stress-resistance effects to prolong the lifespans of yeasts.

## 5. Conclusions

IQ from *A. venetum* L. exerts obvious antiaging effects on yeasts. It extends their lifespans by improving the stress resistances and inducing mitophagy via the Sch9/Rim15/Msn signaling pathway (Figure 9). We will evaluate the efficacy and safety of IQ on mammalian cells and animal models to develop it as a promising candidate that can prevent and treat aging and age-related disorders.

## Figures and Tables

**Figure 1 antioxidants-12-01939-f001:**
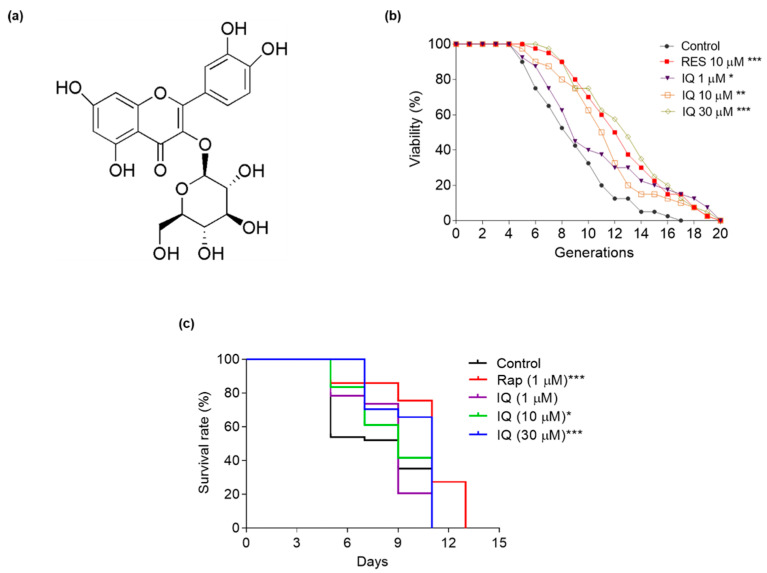
The chemical structure and antiaging effects of isoquercitrin (IQ) on yeasts. (**a**) The chemical structure of IQ. (**b**) Effect of IQ on the replicative lifespan of K6001 yeasts. Resveratrol (RES) at 10 μM was used as the positive control. (**c**) Effect of IQ on the chronological lifespan of YOM36 yeasts. Rapamycin at 1 μM was used as the positive control. *, **, and *** represent significant differences compared to the control group at *p* < 0.05, *p* < 0.01, and *p* < 0.001, respectively.

**Figure 2 antioxidants-12-01939-f002:**
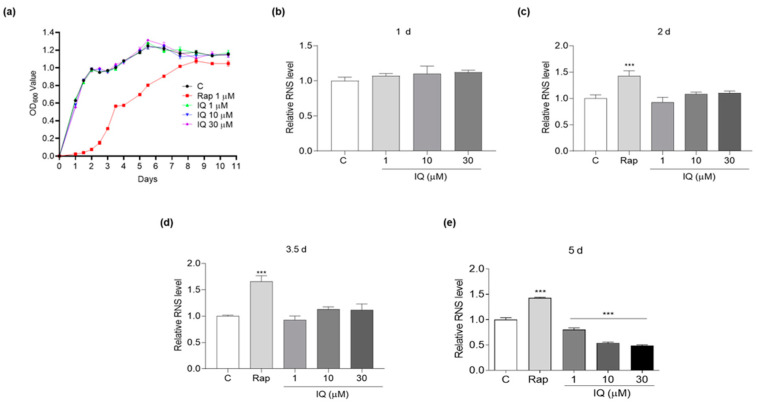
Effect of IQ on reactive nitrogen species (RNS) levels during chronological aging. (**a**) The growth curve of YOM36 yeasts during chronological aging. (**b**–**e**) The changes in the RNS levels after IQ treatment at 1, 2, 3.5, and 5 days. *** indicates significant differences from the control group at *p* < 0.001.

**Figure 3 antioxidants-12-01939-f003:**
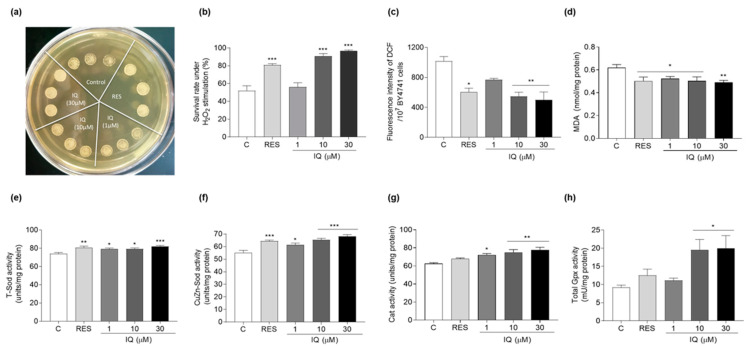
Effect of IQ on the survival of yeasts under oxidative stress, and evaluation of ROS, MDA levels and antioxidant enzyme activity in yeasts. (**a**) The colony formation of BY4741 yeasts upon IQ treatment under H_2_O_2_ stimulation at 11 mM. (**b**) The survival rate of BY4741 after IQ treatment under oxidative stress induced by 5.5 mM H_2_O_2_. (**c**,**d**) Effects of IQ on reactive oxygen species (ROS) and malondialdehyde (MDA) in yeasts under physiological status. (**e**–**h**) Changes in antioxidant enzyme activity in BY4741 yeast after incubation with IQ for 24 h. *, **, and *** indicate significant differences from the control group at *p* < 0.05, *p* < 0.01, and *p* < 0.001, respectively.

**Figure 4 antioxidants-12-01939-f004:**
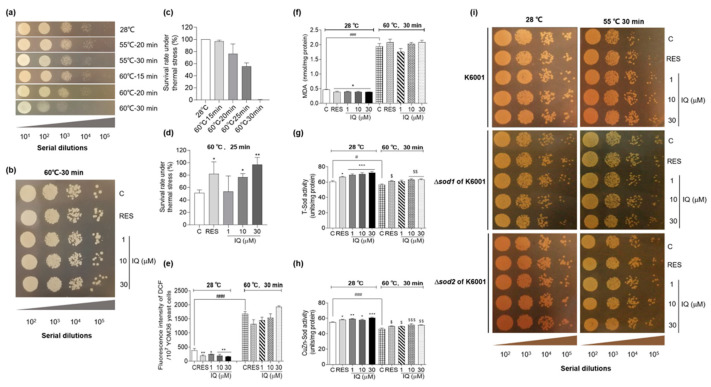
IQ enhances the Sod activity to synergistically counteract thermal stress in yeasts. (**a**) Photograph of YOM36 yeasts with or without thermal stimuli after culturing at 28 °C for 24 h. (**b**) The growth of YOM36 yeasts with IQ treatment after heating at 60 °C for 30 min and culturing at 28 °C for 48 h. (**c**) The survival rate of YOM36 yeasts under different thermal-stress conditions. (**d**) The quantitative anti-thermal-stress experimental results after IQ treatment followed by heating at 60 °C for 25 min. * *p* < 0.05 and ** *p* < 0.01 represent significant differences compared to the control group. (**e**,**f**) The ROS and MDA in YOM36 yeasts after treatment with IQ under room-temperature and thermal-stress conditions. * *p* < 0.05 and ** *p* < 0.01 represent significant differences compared with the control group under room-temperature conditions. ### *p* < 0.001 represents significant difference between the negative control group. (**g**,**h**) The T-Sod and CuZn-Sod activities in YOM36 yeasts after treatment with IQ under room-temperature and thermal-stress conditions. * *p* < 0.05, ** *p* < 0.01, and *** *p* < 0.001 represent significant differences compared with the control group under room-temperature conditions. $ *p* < 0.05, $$ *p* < 0.01, and $$$ *p* < 0.001 represent significant differences compared with the control group under thermal-stress conditions. # *p* < 0.05 and ### *p* < 0.001 represent significant differences between the negative control group. (**i**) The growth of K6001 yeasts and its Δ*sod1* and Δ*sod2* mutants after IQ treatment with or without heating at 55 °C for 30 min.

**Figure 5 antioxidants-12-01939-f005:**
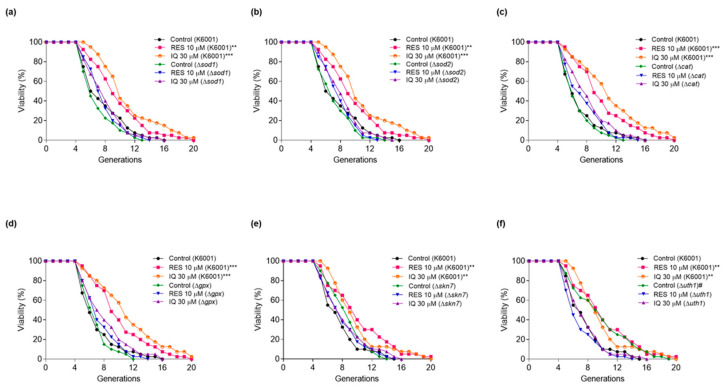
Effect of IQ on the replicative lifespans of Δ*sod1*, Δ*sod2*, Δ*cat*, Δ*gpx*, Δ*skn7*, and Δ*uth1* yeasts with K6001 background. (**a**–**f**) The replicative lifespans of Δ*sod1*, Δ*sod2*, Δ*cat*, Δ*gpx*, Δ*skn7*, and Δ*uth1* yeasts. # *p* < 0.05, ** *p* < 0.01, and *** *p* < 0.001 represent significant differences compared with the negative control group of K6001.

**Figure 6 antioxidants-12-01939-f006:**
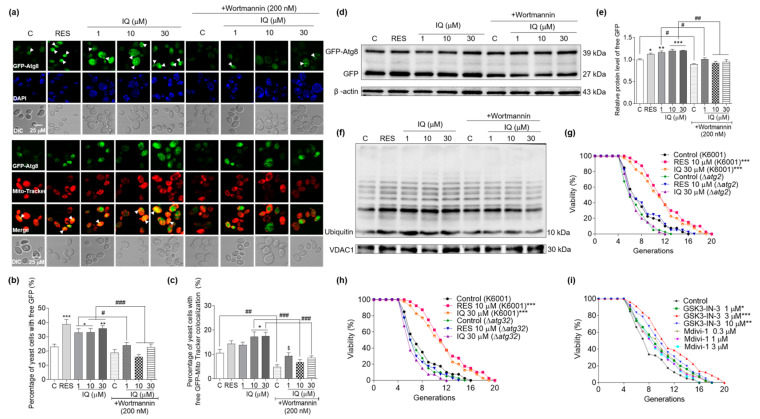
Effect of IQ on autophagy induction in YOM38 yeasts. (**a**) The fluorescent images of autophagy and mitophagy induced by IQ with or without wortmannin inhibition. (**b**) The digitized results of YOM38 yeasts containing free green fluorescent protein (GFP) in (**a**). (**c**) The statistical results of YOM38 yeasts with the colocalization of free GFP (green) and MitoTracker Red CMXRos (red) in (**a**). (**d**) Western blot analysis of GFP-Atg8 and free GFP in YOM38 yeast after treatment with IQ at 1, 10, and 30 μM with or without wortmannin for 22 h. (**e**) The digital results of free GFP in (**d**). (**f**) The changes in ubiquitin in the mitochondria after treatment with IQ with or without wortmannin. (**g**,**h**) The replicative lifespans of Δ*atg2* and Δ*atg32* of K6001 yeasts. (**i**) The replicative lifespans of K6001 yeasts after treatment of mitophagy activator GSK3-IN-3 or inhibitor Mdivi-1. The average lifespans of K6001 and mutants are displayed in Appendix A. * *p* < 0.05, ** *p* < 0.01, and *** *p* < 0.001 represent significant differences compared with the control group without wortmannin. $ *p* < 0.05 represents significant difference compared with the control group with wortmannin. # *p* < 0.05, ## *p* < 0.01, and ### *p* < 0.001 indicate significant differences between the groups with or without wortmannin inhibition.

**Figure 7 antioxidants-12-01939-f007:**
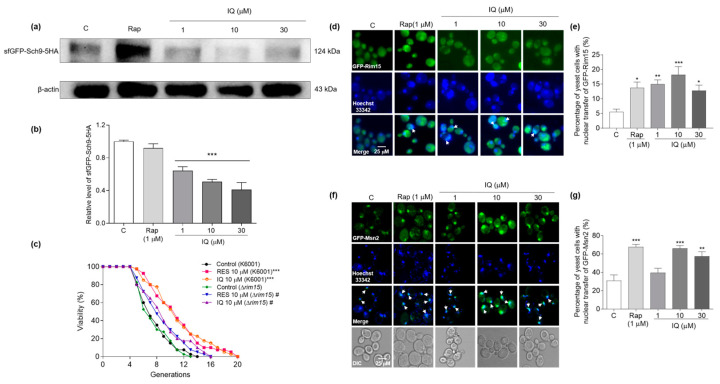
Effect of IQ on the expressions of Sch9 and nuclear translocation of GFP-Rim15 and GFP-Msn2. (**a**) The changes in Sch9 after treatment with IQ for 2 h. (**b**) The digital results of (**a**). (**c**) The replicative lifespans of Δ*rim15* of K6001 after IQ treatment. (**d**) The fluorescence images of the colocalization of GFP-Rim15 and nuclear staining with Hoechst 33,342. (**e**) The statistical results of (**d**). (**f**) The fluorescence signals of the colocalization of GFP-Msn2 and nuclear staining with Hoechst 33,342. (**g**) The statistical results of (**f**). *, **, and *** indicate significant differences from the control group at *p* < 0.05, *p* < 0.01, and *p* < 0.001, respectively. # indicates significant differences from the control group in Δ*rim15* of K6001 at *p* < 0.05.

**Figure 8 antioxidants-12-01939-f008:**
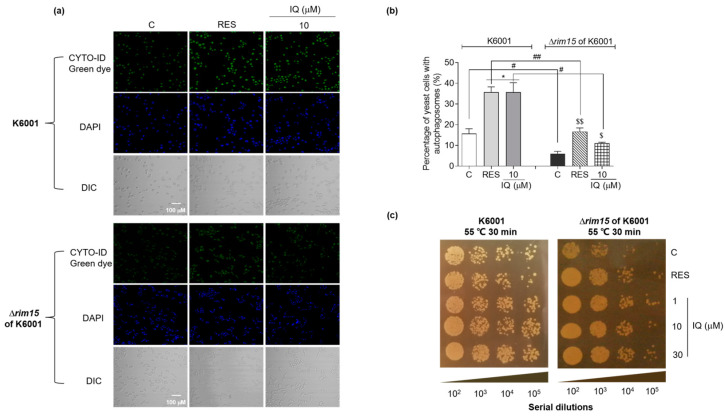
Rim15 is essential for IQ to exert autophagy-induction and thermal-stress-resistance effects. (**a**) The fluorescence images showing the autophagosomes in K6001 and ∆*rim15* of K6001 stained with CYTO-ID green dye. (**b**) The digital results of (**a**). (**c**) The growth of K6001 and ∆*rim15* of K6001 yeasts after IQ treatment with heating at 55 °C for 30 min and culturing at 28 °C for 48 h. * indicates significant differences from the control group of K6001 yeasts at *p* < 0.05. # and ## represent significant differences between the same treatment group of K6001 and ∆*rim15* of K6001 yeasts at *p* < 0.05 and *p* < 0.01. $ and $$ indicate significant differences from the control group of ∆*rim15* of K6001 yeasts at *p* < 0.05 and *p* < 0.01, respectively.

**Figure 9 antioxidants-12-01939-f009:**
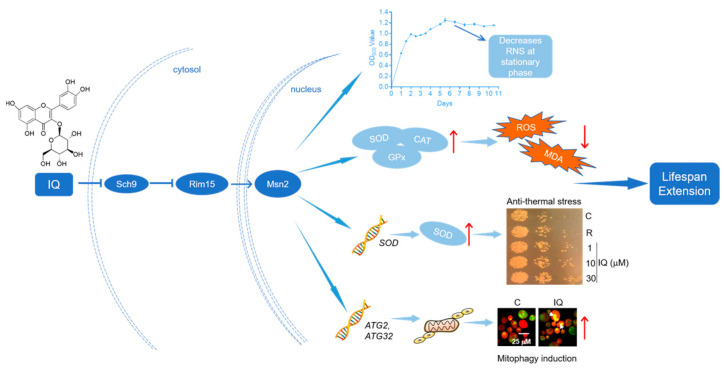
The proposed mechanism of action for IQ. Improvements in stress resistance and mitophagy induction through the Sch9/Rim15/Msn signaling pathway are involved in the antiaging effect of IQ on yeasts.

## Data Availability

All figures and data used to support this study are included within this article.

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
