# Peer review of "Isoquercitrin from Apocynum venetum L. Exerts Antiaging Effects on Yeasts via Stress Resistance Improvement and Mitophagy Induction through the Sch9/Rim15/Msn Signaling Pathway"

_antioxidants, 2023, doi:10.3390/antiox12111939_

Round 1
Reviewer 1 Report
Comments and Suggestions for Authors
The manuscript is interesting. however, needs some major modification
1. Provide more supporting data to activate mitophagy, improving yeast's aging.
2. The discussion part need more scientific discussion with recent citation.
Comments on the Quality of English LanguageNeed minor English language adjustment.
Reviewer 2 Report
Comments and Suggestions for Authors
This is a very thorough and well-done experiment to assess the potential anti-aging effect of a specific plant-derived compound. Each of the individual assays are well-described and include adequate controls. The use of mutant strains to directly implicate specific pathways strengthens the authors conclusions. I have no comments or concerns. Very nicely done.
Author Response
There is no comment from reviewer 2.
Reviewer 3 Report
Comments and Suggestions for Authors
Liu et al. investigated the effect of isoquercitrin (IQ) on yeast cells (K6001 and YOM36).
The manuscript needs major revisions as outlined below:
1. The abstract is not organized in a logical sequence: Background and Aim of the Study, Methods, Results, and Conclusions.
Also, it is necessary to give the reader additional information to understand some sentences. For example: What is sod (line 17) and what is its role? Without this information, the reader cannot understand why the authors discuss superoxide dismutase.
2. The introduction section should be shortened and focus on IQ structure and function, biological activity, differences from other antioxidant molecules.
The role of yeast as an experimental model was adequately discussed. However, some information about YOM36 yeasts should be added.
The signaling pathways responsible for stress resistance should be better described.
3. Why do the Authors use an IQ concentration of 1-30 microM?
4. The figures seem too complex and have a low resolution.
5. The Authors should put more effort into finding out what is new in the present work compared to previous studies and what we can learn from the data.
What are the implications of this study for humans?
Round 2
Reviewer 1 Report
Comments and Suggestions for Authors
The present form of the manuscript is now improved.
Reviewer 3 Report
Comments and Suggestions for Authors
The authors have responded to all my suggestions, and I have no further comments.
In my opinion, the manuscript is now suitable for publication.
PS: In the abstract, the headings can be removed.